

# VBM-YOLO: an enhanced YOLO model with reduced information loss for vehicle body markers detection

Bin Wang, Chao Li, Chao Zhou and Jun Sun

School of Artificial Intelligence and Computer Science, Jiangnan University, Wuxi, Jiangsu, China

## ABSTRACT

In vehicle safety detection, the accurate identification of body markers on medium and large vehicles plays a critical role in ensuring safe road travel. To address the issues of the feature and gradient information loss in previous You Only Look Once (YOLO) series models, a novel Vehicle Body Markers YOLO (VBM-YOLO) model has been designed. Firstly, the model integrates the cross-spatial-channel attention (CSCA) mechanism proposed in this study. The CSCA uses cross-dimensional information to address interaction issues during the fusion of spatial and channel dimensions, significantly enhancing the model's representational capacity. Secondly, we propose a multi-scale selective feature pyramid network (MSSFPN). By a progressive fusion approach and multi-scale feature selection learning, MSSFPN alleviates the issues of feature loss and target layer information confusion caused by traditional top-down and bottom-up feature pyramids. Finally, an auxiliary gradient branch (AGB) is proposed. During training, AGB incorporates feature information from different target layers to help the current layer retain complete gradient information. Additionally, the AGB branch does not participate in model inference, thereby reducing additional overhead. Experimental results demonstrate that VBM-YOLO improves mean average precision (mAP) by 2.3% and 4.3% at intersection over union (IoU) thresholds of 0.5 and 0.5:0.95, respectively, compared to YOLOv8s on the vehicle body markers dataset. VBM-YOLO also achieves a better balance between accuracy and computational resources than other mainstream models, exhibiting good generalization performance on public datasets like PASCAL VOC and D-Fire.

# INTRODUCTION

The recognizability (*Balasubramanian & Bhardwaj, 2018*) and protectiveness (*Kortağ & Göncü, 2021*) of vehicles themselves are crucial to traffic safety. Particularly for medium and large vehicles, their safety while on the road is intimately tied to the presence of their vehicle body markers. The primary vehicle body markers include rear reflective signs, side reflective signs, rear signs, rear guards, and side guards. These markers significantly enhance the visibility and protective capabilities of vehicles under varying visibility

Corresponding authors
Chao Li, lcmeteor@hotmail.com
Jun Sun, sunjun_wx@hotmail.com

conditions and road situations. Specifically, in environments characterized by low visibility at night or due to restricted lines of sight, reflective signs notably enhance the recognizability of vehicles, effectively reducing the risk of collision. For large transport vehicles, the conspicuous design of rear signs effectively prompts following vehicles to maintain a necessary, safe distance, thus preventing rear-end collisions. Furthermore, guard boards can mitigate collision risks and prevent small vehicles from being entrapped under the chassis of medium and large vehicles in the event of an accident. Given the essential role of triangle signs in all vehicles, they are considered an extension of the vehicle body safety markers. In the case of vehicle malfunction requiring parking, a triangle warning sign serves as a temporary safety warning facility, safeguarding the vehicle's and its occupants' safety. Consequently, before permitting normal vehicle operation, relevant authorities should inspect these vehicle body markers to ensure road safety compliance. This process can be effectively achieved using object detection technology based on computer vision.

In recent years, there has been rapid development in the field of object detection. Increasing numbers of researchers are dedicating efforts to applying object detection algorithms in practical scenarios. For instance, *Mittal, Chawla & Tiwari (2023)* trained a hybrid model of Faster R-CNN and YOLO to estimate traffic density. *Liu et al. (2023)* introduced the C3Ghost and GhostConv modules into the backbone network of YOLOv5 addresses the issue of prolonged computation time and suboptimal detection rates often encountered by robots. *Ye et al. (2022)* designed a network using the LFM module for real-time obstacle detection in railway traffic scenes. *Wang et al. (2022)* proposed the LDS-YOLO model for the rapid identification of dead trees in forests, incorporating the SPP module with SoftPool and stacking depth-wise separable convolutions within the network.

While YOLO-series detectors have achieved remarkable progress in general object detection, we identified three critical limitations when applying existing YOLO-based frameworks to vehicle body markers detection: First, crucial targets like reflective signs and rear signs frequently exhibit small-scale characteristics, where conventional YOLO architectures suffered from progressive detail loss during successive downsampling operations. Second, high-density spatial distributions and inter-class similarities among vehicle markers led to feature confusion and missed detections in current implementations. Third, illumination variations in practical scenarios significantly degraded detection robustness, particularly under extreme lighting conditions. To address these challenges, we design the model architecture at the information level and propose the Vehicle Body Markers YOLO (VBM-YOLO) model based on YOLOv8s. The following summarizes the main contributions of this study:

1) A novel cross-spatial-channel attention (CSCA) mechanism is proposed. This mechanism leverages cross-dimensional information to establish a global interdependence between previously unrelated spatial and channel dimensions. This significantly enriches the model's feature information and enhances its detection performance.

2) We propose a multi-scale selective feature pyramid network (MSSFPN). MSSFPN uses a progressive fusion approach that allows features from non-adjacent levels to be directly integrated, mitigating the loss of feature information during transmission. Each target layer in MSSFPN learns the importance of information from other target layers to selectively extract critical information beneficial to the current target layer, thereby avoiding interference from irrelevant information.

3) We propose the auxiliary gradient branch (AGB) to mitigate the issue of substantial information loss about the target during the transmission of initial gradients from a deep network. The AGB ensures that each target layer receives comprehensive gradient information during backpropagation. Additionally, AGB only supervises during training and does not participate in inference, thus reducing extra overhead.

4) The experimental results indicate that, compared to YOLOv8s, the VBM-YOLO model achieves improvements of 2.3% and 4.3% in mAP@0.5 and mAP@50:95, respectively, on the vehicle body markers dataset. Furthermore, the performance on two publicly available datasets demonstrates its superior generalization.

The remainder of this article is organized as follows. The "Related Work" reviews relevant studies. The proposed VBM-YOLO is detailed in the "Proposed Methodology" section. "Experiments and Analysis" presents and discusses the experimental results. Finally, some conclusions and future ideas are provided in the "Conclusion" section.

## RELATED WORK

### Object detection model

Object detection models are primarily categorized: two-stage detectors and one-stage detectors. Classic two-stage detectors include R-CNN (*Girshick et al., 2014*), Fast R-CNN (*Girshick, 2015*), Faster R-CNN (*Ren et al., 2015*), while representative one-stage models include SSD (*Liu et al., 2016*), FCOS (*Tian et al., 2020*), and YOLO series (*Redmon et al., 2016*; *Redmon & Farhadi, 2017, 2018*; *Bochkovskiy, Wang & Liao, 2020*; *Ultralytics, 2022*; *Li et al., 2023*; *Wang, Bochkovskiy & Liao, 2023*).

Two-stage object detection models improve the accuracy and robustness of detection by first generating candidate boxes and then classifying and adjusting these candidates. R-CNN (*Girshick et al., 2014*), one of the earliest two-stage models, introduced convolutional neural networks (CNNs) to the object detection field. *He et al. (2017)* further enhanced this model by adding a branch for predicting object masks, leading to the development of Mask R-CNN. *Cai & Vasconcelos (2018)* proposed Cascade R-CNN, which achieves higher accuracy by cascading multiple detectors. Despite the methods above attaining high accuracy and superior performance, particularly for large objects and complex scenes, the processing of two independent network stages results in slower speeds.

In contrast, one-stage models directly predict object detection results from input data without undergoing multi-stage processing, thus achieving greater efficiency. OverFeat (*Sermanet et al., 2013*) was one of the earliest CNN-based one-stage models. Currently, RT-DETR (*Zhao et al., 2023*) is regarded as one of the most generalized and accurate one-stage object detection models in the DETR series. However, its extensive parameters

and substantial training resource requirements hinder practical deployment. The YOLO series has been widely adopted in industrial applications due to its real-time detection capabilities. Among them, YOLOv8 (*Terven, Córdova-Esparza & Romero-González, 2023*) is recognized as one of the most advanced versions in the YOLO family. Through continuous iterations, recent versions including YOLOv9 (*Wang, Yeh & Liao, 2024*), YOLOv10 (*Wang et al., 2024*), YOLOv11 (*Khanam & Hussain, 2024*), and YOLOv12 (*Tian, Ye & Doermann, 2025*) have achieved significant improvements in both detection accuracy and efficiency while preserving real-time performance.

Although the YOLO series performs well on large-scale datasets, it may not achieve high precision in specific scenarios, often requiring increased computational resources to enhance accuracy. For the specific task of vehicle body markers detection, we propose the VBM-YOLO model to achieve a balance between accuracy and computational efficiency, maintaining high detection accuracy while using fewer computational resources than similar models.

## Attention mechanisms

In human perception, attention mechanisms involve selectively focusing on relevant information, allowing individuals to disregard irrelevant stimuli. This process helps the brain capture crucial information and grasp essential details. In recent years, various research methods have effectively integrated this attention mechanism into the architecture of deep CNNs to enhance the performance of large-scale visual tasks.

Early work, such as SENet (*Hu, Shen & Sun, 2018*), focused on modeling channel relationships in feature maps by learning weights for each channel. Subsequently, the convolutional block attention module (CBAM) (*Woo et al., 2018*) was proposed. It enriches attention maps by extracting channel and spatial features through max-pooling and average-pooling. To establish connections within the dimensions, DANet (*Fu et al., 2019*) modeled global dependencies of channels and spatial dimensions through self-attention mechanisms. The convolutional triplet attention module (CTAM) (*Misra et al., 2021*) models cross-dimensional dependencies between spatial and channels, highlighting the importance of capturing cross-dimensional dependencies. The coordinate attention (CA) (*Hou, Zhou & Feng, 2021*) addressed the issue of positional information loss in 2D pooling by decomposing channel attention into two 1D encoding processes. Recently, the expectation-maximization attention (EMA) (*Ouyang et al., 2023*) further enhanced the CA module by improving the sequential processing method, avoiding the side effects of channel dimension reduction modeling, and achieving channel reshaping and cross-dimensional fusion.

However, most of the methods mentioned above have significant drawbacks. The majority of attention modules consider only variations within a single dimension. Although modules like CBAM and DANet account for both dimensions, they fail to leverage the interaction between the two dimensions to facilitate better fusion. Similarly, while CTAM considers cross-dimensional information, it does not establish a global dependency within the spatial and channel dimensions, leading to the loss of some critical information. In contrast, our CSCA attention module not only employs self-attention to

model local and global semantic dependencies between spatial and channel dimensions but also achieves information interaction between the two dimensions.

## Pyramid networks

In object detection, researchers have extensively studied the fusion of multi-level features. These approaches aim to enhance detection performance by integrating high-level and low-level information. The well-known feature pyramid network (FPN) (*Lin et al., 2017*) is a classic feature fusion method in deep learning-based computer vision. It utilizes a top-down pathway to merge features at each level, thereby improving the detection performance of objects of varying sizes.

Given the exceptional performance of FPN, several variants have been proposed for different scenarios, such as Path Aggregation Network (PANet) (*Liu et al., 2018*), Bi-directional Feature Pyramid Network (BiFPN) (*Tan, Pang & Le, 2020*), Asymptotic Feature Pyramid Network (AFPN) (*Yang et al., 2023*), Info-FPN (*Chen et al., 2023*), and Attentional Bidirectional Feature Pyramid Network (ABFPN) (*Zeng et al., 2022*). Each variant optimizes the fusion pathways of FPN in distinct ways. However, pyramid networks with up-and-down varying structures, like those mentioned above, require high-level or low-level features to pass through multiple intermediate-scale transformations. This process gradually degrades feature information. Moreover, mixing multiple target features within the network makes extracting information for each target layer challenging. To this end, we propose MSSFPN, a pyramid network with a clear hierarchy that directly integrates low-level and high-level information in a progressive manner, reducing the loss caused by intermediate transformations.

## Deep supervision

Deep supervision has been widely applied to enhance the performance of deep CNNs. These techniques (*Wang et al., 2015*; *Szegedy et al., 2015*; *Wu, Hong & Chanussot, 2022*) typically involve adding auxiliary prediction layers to certain intermediate hidden layers of deep neural networks, thereby supervising the main network branches. This approach helps addresses issues such as gradient vanishing and slow convergence during the training of deep neural networks. Recent studies in segmentation networks (*Qin et al., 2020*; *Qi, Wu & Chan, 2023*) and object detection *Wang, Yeh & Liao (2024)* have also demonstrated that deep supervision mechanisms significantly improve network performance. However, in traditional deep supervision, if shallow layer supervision fails to retain critical information during training, it may lead to error accumulation can occur, hindering subsequent layers from retrieving necessary information. Furthermore, most deep supervision methods require participation throughout the training and inference phases, which may result in additional network overhead. To address these issues, we propose the AGB method, which enables each target layer to mitigate the problem of gradient information vanishing while avoiding extra inference overhead.

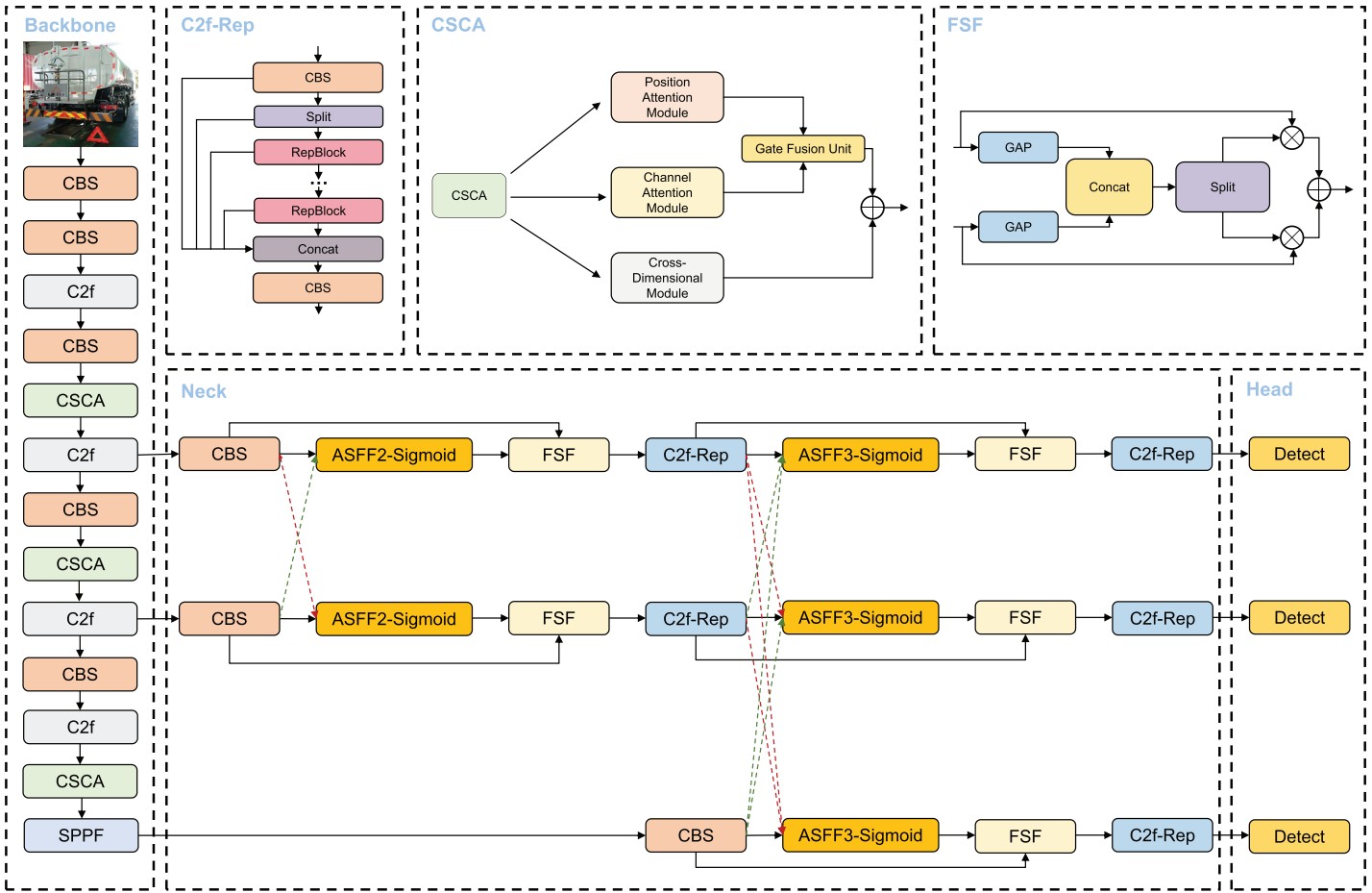

**Figure 1** **The inference architecture of VBM-YOLO. The model includes the proposed CSCA and MSSFPN (Neck).** In the Neck, green dashed arrows represent the process of spatial upsampling and channel dimensionality reduction, while red dashed arrows indicate spatial downsampling and channel dimensionality expansion. In the C2f-Rep, the RepViT Block is abbreviated as RepBlock. The Detect modules in the Head use decoupled heads.

## PROPOSED METHODOLOGY

### Overall architecture

To address the issues mentioned above, we propose a novel VBM-YOLO model for vehicle body markers detection. The inference architecture of VBM-YOLO is shown in Fig. 1. We introduced the CSCA mechanism, which employs self-attention to establish cross-spatial-channel attention within the backbone network. The MSSFPN replaces the original neck network. Furthermore, within the MSSFPN network, we utilized the improved C2f-Rep module to enhance the extraction of channel and spatial information. The designed AGB is incorporated during training to monitor and guide information flow during training, ensuring more effective supervision, as detailed in the "Auxiliary Gradient Branch" section.

The main modules utilized in the VBM-YOLO backbone network, including CBS, C2f, and SPPF. Moreover, like YOLOv8, VBM-YOLO employs decoupled heads. The CBS module comprises three submodules: convolution, batch normalization, and the SiLU

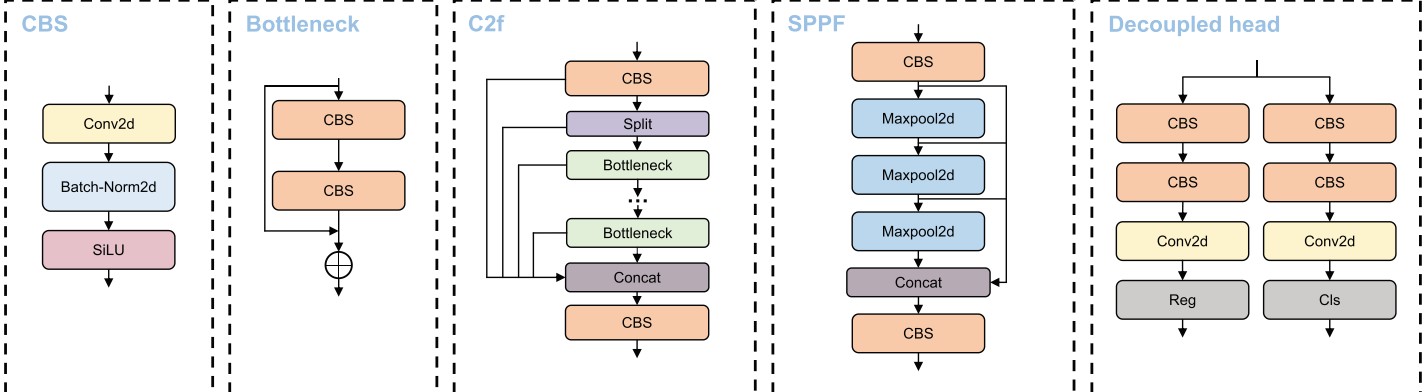

**Figure 2 CBS, C2f, SPPF and decoupled head structure diagram.**

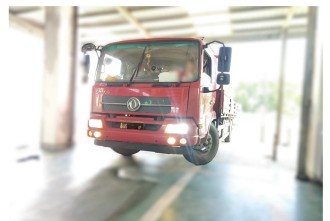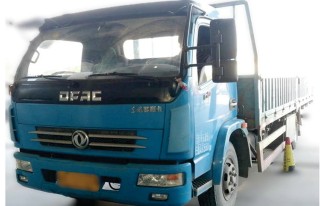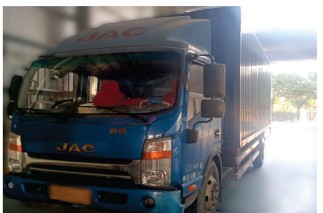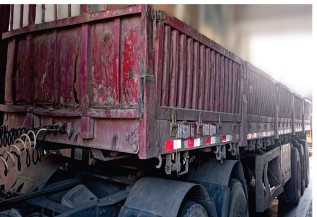

**Figure 3 Images under strong lighting conditions and at varying distances.**

activation function. The C2f module evenly splits the extracted features along the channel direction using a slicing operation. These split features undergo processing through concatenated Bottleneck modules, followed by cross-layer connections. The SPPF module is an efficient pooling module designed to extract and fuse high-level features. This module utilizes repeated max-pooling to extract as many high-level semantic features as possible during the fusion process. Each detection head has two branches: a regression branch and a classification branch, which perform regression and classification tasks, respectively. Each branch consists of two 3 × 3 CBS and a 1 × 1 2D convolution. The detection head module employs an anchor-free approach and a dynamic TaskAlignedAssigner for positive and negative sample allocation. The primary structures of these modules are illustrated in Fig. 2.

## Cross-spatial-channel attention

Vehicles may not always be detected from the optimal angle. Especially in scenarios like those shown in Fig. 3, strong lighting and long distances can make vehicle body markers inconspicuous in the entire image. Additionally, convolution operations produce local receptive fields, which can cause features of targets with the same pixels to differ. Therefore, we establish global dependencies between features through the cross-spatial-channel attention (CSCA) mechanism, adaptively aggregating remote context information, highlighting inconspicuous object features, and addressing feature discrepancy issues.

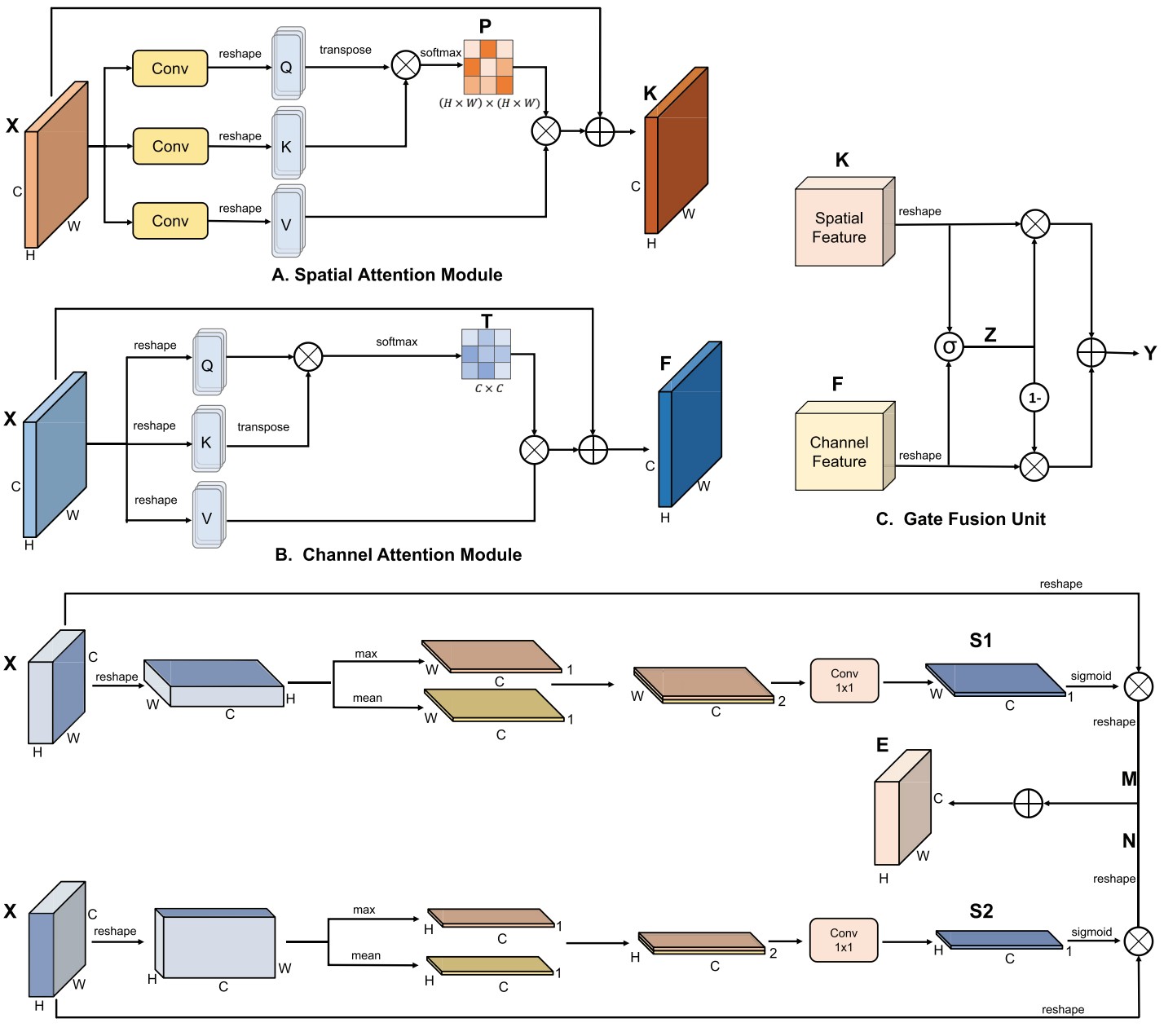

**Figure 4 The different key modules included in the cross-space channel attention (CSCA) mechanism.** As shown in the figure, (A) the spatial attention module, (B) the channel attention module, (C) the gate fusion unit, and (D) the cross-dimensional module. Above is a detailed breakdown of each module.

As shown in Fig. 1, CSCA includes a spatial attention module, a channel attention module, a cross-dimensional module, and a gated fusion unit. CSCA first reduces the dimensionality of features and inputs them into the spatial attention module. The spatial attention module selectively aggregates features at each position through a weighted sum of features at all positions. This ensures that similar features are correlated regardless of

their distance. Meanwhile, the channel attention module selectively emphasizes interdependent channel maps by integrating related features across all channel maps.

Specifically, as shown in Fig. 4, the spatial attention module first generates a spatial attention matrix that models the spatial relationships between any two pixels of the features. Next, matrix multiplication is performed between the attention matrix and the original features. Finally, element-wise summation is conducted between the resulting matrix and the original features to obtain the final representation reflecting remote context. The process of the channel attention module is similar to the spatial attention module, except that the first step is to calculate the channel attention matrix along the channel dimension. We use the gated fusion unit to reallocate the global feature information extracted by the spatial and channel attention modules, obtaining the critical global information that is most beneficial for each target layer. To achieve better feature representation across the spatial and channel dimensions, we fuse the interactive feature information generated by the cross-dimensional module with the gated fusion unit's output through addition and convolution operations. We introduce a CSCA before each target layer to enhance its feature information.

### Spatial attention module

In the YOLO backbone, the features generated due to the limitations of convolution can lead to deviations in object classification and localization. To address this, we introduce a positional attention module, which captures rich contextual relationships within the local features and enhances the model's representational capability.

As shown in Fig. 4A, given a local feature $X \in \mathbb{R}^{C \times H \times W}$, the feature is first passed through three convolutional layers and reshaped to generate the feature maps $Q$, $K$, and $V$, where $\{Q, K, V\} \in \mathbb{R}^{C \times N}$ and $N = H \times W$. Then, the transpose of $Q$ is multiplied by $K$, and a softmax function is applied to compute the spatial attention map $P \in \mathbb{R}^{N \times N}$. Simultaneously, $V$ is multiplied by $P$, and the result is reshaped back to $\mathbb{R}^{C \times H \times W}$. This output is then summed element-wise with the original feature $X$ to obtain the final output $K \in \mathbb{R}^{C \times H \times W}$:

$$P_{ji} = \frac{\exp(Q_i \cdot K_j)}{\sum_{i=1}^{N} \exp(Q_i \cdot K_j)} \tag{1}$$

$$K_j = \sum_{i=1}^{N} (P_{ji} \cdot V_i) + X_j \tag{2}$$

where $P_{ji}$ measures the influence of the $i^{th}$ position on the $j^{th}$ position. The greater the similarity of the feature representations, the higher the correlation. $K_j$ represents the weighted sum of the features from all other positions and the original feature at the $j^{th}$ position.

### Channel attention module

Each channel map in the high-level features can be viewed as a response to a specific category. The semantic information generated within these maps is interrelated. By

modeling the relationships among these channel feature maps, we can highlight relevant feature maps and strengthen the connections between related semantics.

Unlike spatial attention modules, as illustrated in Fig. 4B, the channel attention module directly reshapes the original features $X \in \mathbb{R}^{C \times H \times W}$ to generate three feature maps $Q$, $K$, and $V$, all of which are of size $\mathbb{R}^{C \times N}$. Subsequently, $Q$ is multiplied by the transpose of $K$, followed by a softmax operation to compute the channel attention map $T \in \mathbb{R}^{C \times C}$. Then, a matrix multiplication is performed between $T$ and $V$, reshaping the resultant to $\mathbb{R}^{C \times H \times W}$, which is summed element-wise with $X$ to yield the final output $F \in \mathbb{R}^{C \times H \times W}$:

$$T_{ji} = \frac{\exp(Q_i \cdot K_j)}{\sum_{i=1}^{C} \exp(Q_i \cdot K_j)} \tag{3}$$

$$F_j = \sum_{i=1}^{C} (T_{ji} \cdot V_i) + X_j \tag{4}$$

where $T_{ji}$ quantifies the influence of the $i^{th}$ channel on the $j^{th}$ channel, and $F_j$ represents the weighted sum of the features from all other channels and the original feature at the $j^{th}$ channel.

### Gated fusion unit

As illustrated in Fig. 4C, a gated fusion unit is designed to selectively filter and capture essential spatial and channel information of the current target layer. Specifically, the spatial features $K$ and channel features $F$ are reshaped to $\mathbb{R}^{H \times W \times C}$. The reshaped features are then processed through linear layers and summed. Subsequently, a sigmoid function is applied to the summed features to generate a weight representation $Z \in \mathbb{R}^{H \times W \times C}$. The weight representation $Z$ and its complement $1 - Z$ are then used to scale the reshaped $K$ and $F$, respectively. The scaled features are summed and reshaped to produce the fused features $Y \in \mathbb{R}^{C \times H \times W}$. The calculations are as follows:

$$Z = \frac{e^{(KW_1 + FW_2)}}{e^{(KW_1 + FW_2)} + 1} \tag{5}$$

$$Y = Z \cdot K + (1 - Z) \cdot F \tag{6}$$

where $W_1$ and $W_2$ are parameter matrices used for generating the weights. This process ensures the effective fusion of spatial and channel features, thereby obtaining a more precise feature representation.

### Cross-dimensional module

Since the spatial and channel modules are separated when establishing contextual information, a simple addition fusion operation without additional information cannot fully represent the contextual information. Therefore, the interactive information between the two dimensions is necessary. To this end, we introduce a cross-dimensional module designed to capture the interaction between spatial and channel dimensions. First, we enable interaction between the channel and the spatial dimensions $H$ and $W$. Next, the generated information is multiplied by the original feature information and then added together. The final result represents the interactive information between the channel and

spatial dimensions. As illustrated in the upper part of Fig. 4D, the feature $X \in \mathbb{R}^{C \times H \times W}$ is first reshaped, then subjected to max pooling and average pooling along the $H$ spatial dimension, resulting in two effective features of size $\mathbb{R}^{1 \times C \times W}$. These features are concatenated along the $H$ spatial dimension and then passed through a $1 \times 1$ convolutional layer to obtain the weight information $S1 \in \mathbb{R}^{1 \times C \times W}$ related to the channel and the $H$ spatial dimension. The weight information is multiplied with the original features after passing through a sigmoid function, resulting in features that, after reshaping, become $M \in \mathbb{R}^{C \times H \times W}$. $M \in \mathbb{R}^{C \times H \times W}$ represents the interaction between the channel and the $H$ spatial dimension.

As shown in the lower part of Fig. 4D, a similar process is performed along the $W$ spatial dimension to obtain the feature $N \in \mathbb{R}^{C \times H \times W}$, representing the interaction between the channel and the $W$ spatial dimension. Finally, the two features $M$ and $N$ are added together to obtain the complete interaction feature $E \in \mathbb{R}^{C \times H \times W}$:

$$S1 = f_1\left(\max(X)_H, \mathrm{mean}(X)_H\right) \tag{7}$$

$$S2 = f_2\left(\max(X)_W, \mathrm{mean}(X)_W\right) \tag{8}$$

$$M = X \cdot \frac{e^{S1}}{e^{S1} + 1} \tag{9}$$

$$N = X \cdot \frac{e^{S2}}{e^{S2} + 1} \tag{10}$$

$$E = M + N \tag{11}$$

where $\max(X)_H$ represents the max pooling of the feature $X$ along the $H$ dimension, $\mathrm{mean}(X)_H$ represents the average pooling along the $H$ dimension, and $f$ denotes the series of concatenation and convolution operations.

## Multi-scale selective feature pyramid network

To address the confusion and degradation of feature information caused by the previous pyramid structure, we propose the multi-scale selective FPN, as illustrated in the neck section of Fig. 1. MSSFPN employs a clear hierarchical structure for each target layer to avoid confusion. Specifically, it first fuses the low-level information from two target layers and then progressively integrates the higher-level information from deeper layers. This approach prevents information loss associated with top-down operations and mitigates conflicts from directly fusing low-level and high-level information with significant semantic gaps. Direct summation during feature fusion inevitably leads to information loss. Therefore, we introduce the ASFF (*Liu, Huang & Wang, 2019*) module to learn feature information from different target layers adaptively. Although ASFF handles the fusion of features at different scales, preserving the crucial feature information of the current layer is also essential. Thus, we propose the feature selective fusion (FSF) module, which selectively integrates original and multi-scale information better to capture the crucial information of the current target layer. In the FSF module, we first perform global average pooling on both the original and fused features to obtain two sets of pooled feature information. These sets are then concatenated along the channel dimension and processed using softmax and sigmoid, respectively, to generate two sets of weights representing the

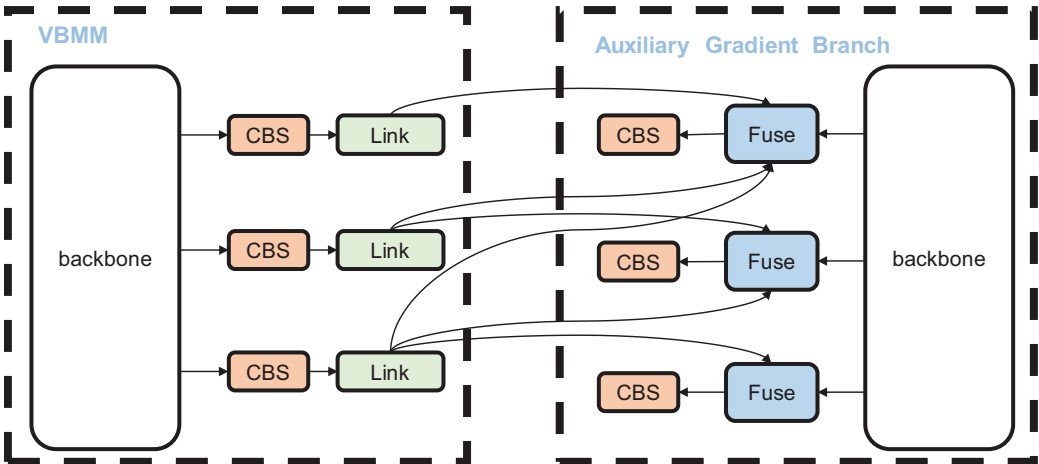

**Figure 5** **The structural diagram of the model during training.** The auxiliary gradient branch (AGB) is utilized to supervise the gradient backpropagation of the VBM-YOLO during training.

importance levels for the current target layer. These weights are multiplied with the original features, and the resulting features are fused to produce information beneficial for the current layer.

Additionally, our research finds that the BottleNeck module in the original C2f structure of YOLOv8, which uses two consecutive 3 × 3 convolutions, leads to the loss of some channel fusion information. Consequently, within the MSSFPN, we replace the BottleNeck module with the RepViT Block module from RepViT (*Wang et al., 2023*) in C2f. The RepViT Block module, with its multi-branch structure and 1 × 1 and 3 × 3 depthwise convolutions, retains channel information and enhances spatial information while having fewer parameters than the original BottleNeck module. Furthermore, we utilize the reparameterization technique from the RepViT Block to reduce GFLOPs by 1.3 during inference.

## Auxiliary gradient branch

In object detection algorithms, information from each target layer is transmitted from the preceding layer to the current one, often resulting in information loss. This issue becomes more pronounced as the network depth increases. Therefore, it is crucial to maintain the integrity of each target layer's information, which requires the complete transmission of information from the previous layer to the current one, both in forward and backward propagation. Given our improvements to the network structure, we aim to preserve the gradient information of each target layer from the perspective of backward propagation. As illustrated in Fig. 5, to ensure consistency in information during forward and backward propagation, the backbone structure in the AGB is kept consistent with the backbone in VBM-YOLO. Each target layer in VBM-YOLO is connected to all corresponding preceding target layers in the AGB, enabling it to receive gradient information from the previous layer and thus maintain its information integrity. For instance, during training, to ensure the completeness of gradient information in the third target layer of VBM-YOLO,

the third target layer is sequentially connected to the first to third target layers of the AGB branch. During backpropagation, the gradient information from the first to third layers in the AGB is sequentially propagated to the third target layer of VBM-YOLO, thus ensuring the integrity of the third target layer's information.

When employing the AGB, an additional branch is added behind each CBS module in VBM-YOLO to connect a Link module. The Link module, composed of $1 \times 1$ convolutions, sequentially converts the input feature maps to match the channel number of the target layers in the AGB. These converted feature maps are then sequentially assigned to the Fuse module in the AGB for merging. The Fuse module upsamples the features from different target layers and sums them to obtain fusion information on different layers. Moreover, the AGB is used only during training and removed during inference to reduce computational overhead.

## EXPERIMENTS AND ANALYSIS

### Datasets

This study employs portable mobile devices to collect vehicle images requiring body marking detection. From an extensive image pool, we curated a representative dataset containing 7,164 images-comprising 3,582 right-rear 45° vehicle views and 3,582 left-front 45° vehicle views. As shown in Fig. 3, to better simulate real-world detection environments and enhance model robustness, the dataset intentionally includes 122 images with significant overexposure and 132 images captured from substantially long distances. As shown in Fig. 6, this dataset contains six detection targets: rear reflective signs, side reflective signs, rear guards, side guards, rear signs, and triangle signs. Specifically, the rear right side images contain rear reflective signs, side reflective signs, rear guards, side guards, rear signs, and triangle signs. The front left side images contain side reflective signs and side guards.

Ground truth annotations were generated using the LabelImg software. The dataset was divided into a training set of 6,164 images and a validation set of 1,000 images, with an equal 1:1 distribution of right rear and left front images in both sets. Figure 7 illustrates the distribution of labels in the vehicle body markers dataset used in this study. To further verify the generalization of our model across different tasks, we conducted tests on two public datasets: the PASCAL VOC (*Everingham et al., 2010*) dataset and the D-Fire (*de Venâncio, Lisboa & Barbosa, 2022*) dataset, as shown in Fig. 7 for label distribution. The PASCAL VOC dataset, including the 2007 and 2012 versions, contains 20 categories of detection objects. In our experiments, we used 16,551 images from the training and validation sets of VOC2007 and VOC2012 for training and 4,952 images from the VOC2007 test set for validation. The D-Fire dataset, which consists of images of fire and smoke events, has two categories: smoke and fire, with 21,527 images. We used the official split of 17,221 images for training and 4,306 images for validation.

### Experimental setup

The experiments were conducted on an NVIDIA GeForce RTX 4090 (24 GB) GPU, utilizing the environmental dependencies of PyTorch 2.0.1, Python 3.9.17, and CUDA

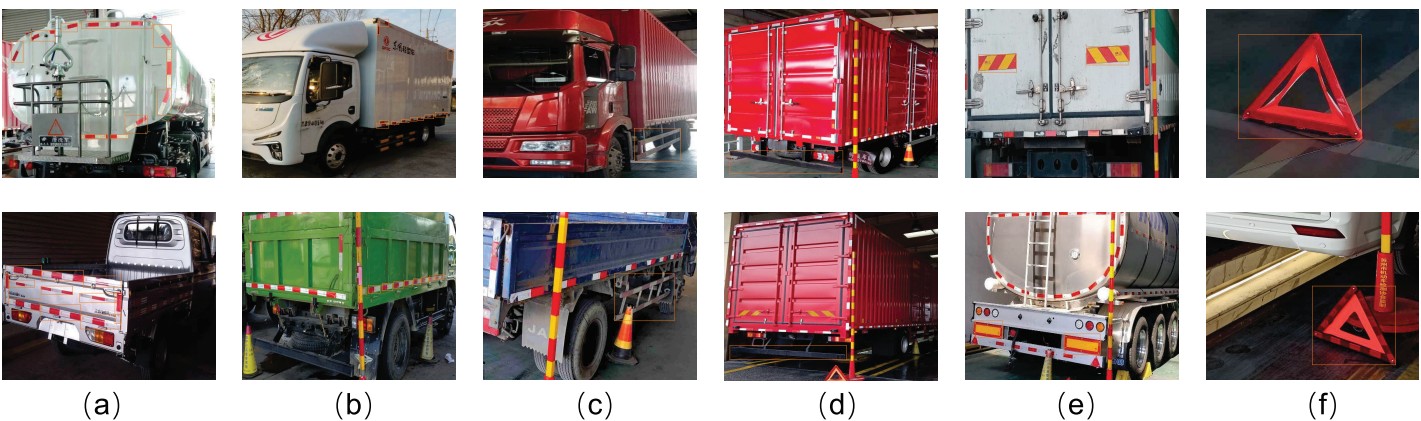

**Figure 6 Different types of vehicle body markers are framed in the figures.** As shown in the figure, (A) rear reflective signs, (B) side reflective signs, (C) side guards, (D) rear guards, (E) rear signs, and (F) triangle signs.

**Figure 7 Label distribution across different datasets.** As shown in the figure, (A) represents the vehicle body markers dataset, (B) represents the VOC dataset, and (C) represents the D-Fire dataset.

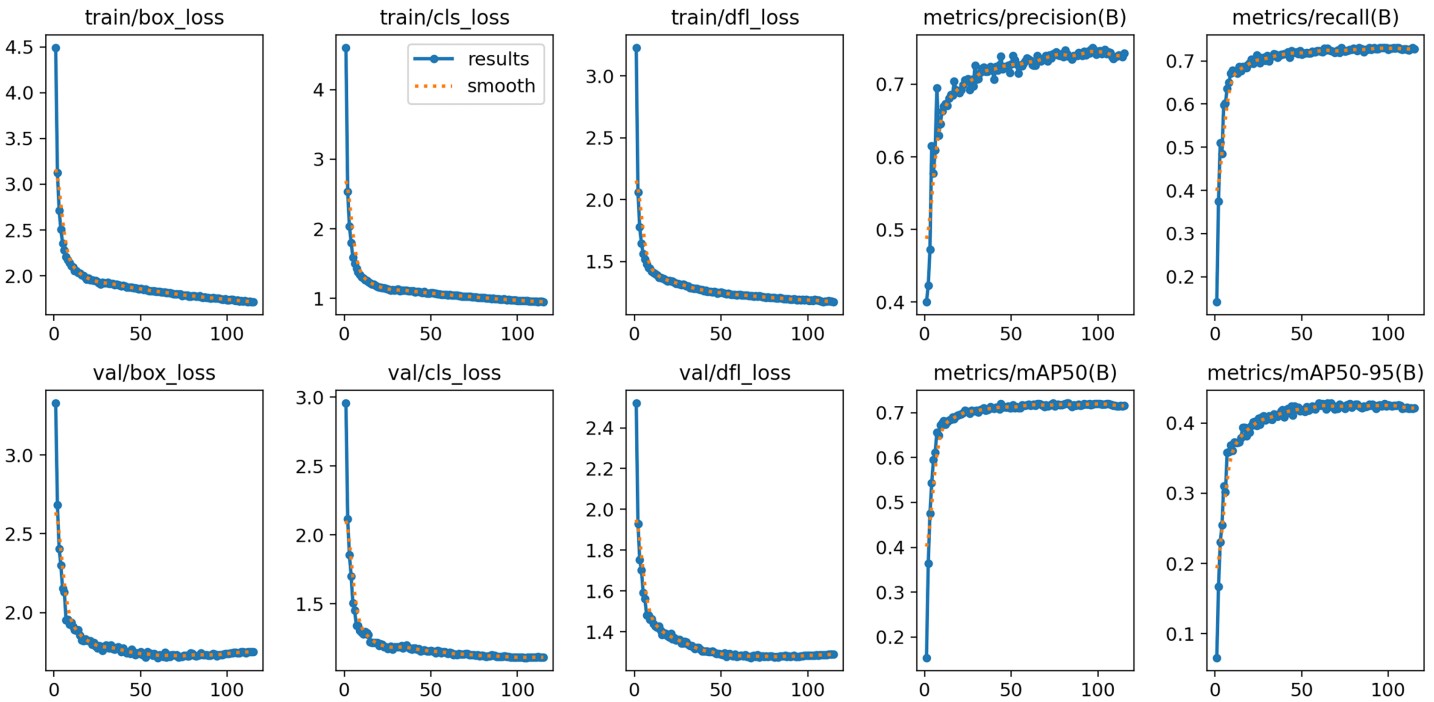

**Figure 8 Evolution of evaluation indexes of YOLOv8s model during training based on the vehicle body markers dataset.**

11.7. During training, we used the SGD optimizer with a batch size of 16. The hyperparameters for SGD were set as follows: momentum at 0.937, initial learning rate at 0.01, and weight decay at 0.0005. All input images were uniformly scaled to $640 \times 640$ pixels. As depicted in Fig. 8, the training process of the YOLOv8s model for our collected vehicle body markers dataset exhibited stability in all evaluation metrics after 125 to 200 epochs. Therefore, the maximum epoch count was set to 200 for models trained using our collected dataset. For the PASCAL VOC dataset and the D-Fire dataset, the maximum epoch count was set to 300 to ensure adequate training and optimal performance on these datasets. In this study, all models were trained from scratch without using pre-trained weights.

## Data augmentation

To enhance the model's generalization capability, we systematically applied a range of data augmentation techniques. We introduced the Mosaic augmentation method, which generated composite training samples by randomly selecting, stitching, and cropping four images. This approach not only effectively improved the detector's adaptability to various backgrounds through multi-background fusion but also allowed the Batch Normalization (BN) layers to learn the feature distributions of four images simultaneously within a single batch, thereby significantly enhancing the model's feature extraction performance.

Regarding the fundamental augmentation strategies, we employed a variety of multi-dimensional geometric transformations: (1) mirror flipping, which produced

mirrored samples *via* horizontal and vertical flips to increase the diversity of the training data; (2) orthogonal rotation, which applied rotations in multiples of 90° to enhance the model's resilience to changes in object orientation; (3) random cropping, which involved sampling local regions to force the model to focus on the most salient features of the targets; and (4) angle rotation, which performed rotations at arbitrary angles in combination with a dynamic bounding box coordinates correction algorithm, ensuring precise spatial alignment of the detection boxes with the rotated images. The integration of these methods significantly improved the model's robustness against variations in target scale, orientation, and background.

## Evaluation metrics

In this study, the model size is evaluated based on its number of parameters. Precision ($P$), recall ($R$), and mean average precision ($mAP$) were employed to evaluate the detection performance of the proposed model. $P$ indicates the probability that a detected object is correctly identified among all detected objects. $R$ denotes the probability of correctly identifying positive samples among all positive samples. Average precision ($AP$) measures the accuracy for a specific category and is determined by the area under the precision-recall ($P$–$R$) curve. $mAP$ is the mean of the ($AP$) values for all categories, indicating the overall accuracy of the model. A higher $mAP$ indicates more accurate detection outcomes. The following equations provide detailed explanations of these metrics:

$$P = \frac{TP}{TP + FP} \tag{12}$$

$$R = \frac{TP}{TP + FN} \tag{13}$$

$$AP = \int_0^1 P(R)dR \tag{14}$$

$$mAP = \frac{\sum_{n=1}^{N} AP(n)}{N} \tag{15}$$

where $TP$ denotes the number of true positive samples, $FP$ denotes the number of false positive samples, and $FN$ denotes the number of false negative samples. $N$ represents the number of classes in the dataset.

Furthermore, $mAP@0.5$ refers to the mean average precision when the Intersection over Union (IoU) threshold is set to 0.5, while $mAP@0.5 : 0.95$ indicates the mean average precision across IoU thresholds ranging from 0.5 to 0.95 with increments of 0.05.

## Comparison experiments

In this section, we conduct experiments on three datasets and compare VBM-YOLO with mainstream one-stage and two-stage object detection models to evaluate its effectiveness.

Table 1 presents the performance comparison between current mainstream models and our proposed VBM-YOLO model on the vehicle body markers dataset. The proposed VBM-YOLO model exhibits superior performance compared to other models. Compared with two-stage models such as Faster R-CNN and SSD, VBM-YOLO shows significant advantages in model size and mAP. Although our VBM-YOLO model is larger than the

**Table 1 Comparison of different object detection models on the vehicle body markers dataset.**

| Model | mAP@0.5 | mAP@0.5:0.95 | Params (M) | GFLOPs | FPS |
|---|---|---|---|---|---|
| SSD | 0.618 | 0.325 | 24.1 | 30.6 | 132 |
| Faster-RCNN | 0.681 | 0.396 | 136.8 | 370.2 | 17 |
| YOLOv5m | 0.726 | 0.434 | 25.1 | 48.3 | 112 |
| YOLOv5s | 0.721 | 0.411 | 7.0 | 15.8 | 140 |
| YOLOv6s | 0.723 | 0.431 | 16.3 | 45.3 | 162 |
| YOLOv7 | 0.734 | 0.423 | 37.2 | 103.2 | 118 |
| YOLOv7-tiny | 0.719 | 0.404 | 6.0 | 13.1 | 165 |
| YOLOv8s | 0.721 | 0.428 | 11.1 | 28.4 | 145 |
| YOLOv8m | 0.731 | 0.449 | 25.9 | 78.9 | 122 |
| RT-DETR-R18 | 0.725 | 0.437 | 20.0 | 60.0 | 85 |
| YOLOv9-C | 0.73 | 0.447 | 25.4 | 102.1 | 101 |
| Gelan-C | 0.729 | 0.446 | 25.4 | 102.1 | 101 |
| YOLOv10-M | 0.707 | 0.42 | 16.4 | 59.1 | 110 |
| YOLOv11-M | 0.723 | 0.421 | 20.1 | 68.0 | 95 |
| YOLOv12-M | 0.728 | 0.435 | 20.2 | 67.5 | 76 |
| VBM-YOLO | 0.744 | 0.471 | 18.2 | 44.9 | 135 |

original YOLOv8s, it achieves a notable improvement in mAP, with increases of 2.3% in mAP@0.5 and 4.3% in mAP@0.5:0.9. Moreover, in comparison with the larger YOLOv8m model, VBM-YOLO not only maintains a smaller size but also attains higher mAP@0.5 and mAP@0.5:0.9, with gains of 1.3% and 2.2%, respectively. The table shows that, while YOLOv7-tiny has the smallest number of parameters, its accuracy is substantially lower compared to the VBM-YOLO model. We also compared our model with some of the latest object detection models of similar sizes, including RT-DETR based on the ResNet18 backbone network, YOLOv9-C, and GELAN-C. It should be noted that GELAN-C is an efficient layer aggregation network proposed in YOLOv9. However, compared to YOLOv9-C, which has the highest accuracy and the largest size among the three, our VBM-YOLO model reduces the number of parameters by 28% while still improving mAP@0.5 and mAP@0.5:0.9 by 1.4% and 2.4%, respectively. Comparative evaluations with recently released YOLOv10, YOLOv11, and YOLOv12 demonstrate VBM-YOLO's mAP@0.5 improvements of 3.7%, 2.1%, and 1.6%, and mAP@0.5:0.9 gains of 5.1%, 5.0%, and 3.6% respectively, while maintaining substantially lower computational costs and higher FPS. These results in the table indicate that our model achieves the best balance between accuracy and resource consumption compared to other models. The PR curve in Fig. 9 further illustrates the improved detection performance of VBM-YOLO compared to YOLOv8s across various datasets. Despite the challenges posed by intense lighting and varying distances, which reduce detection accuracy for side reflective signs, our model still outperforms YOLOv8s by 1.4% in detecting these markers. The results from the other two datasets also clearly demonstrate that VBM-YOLO outperforms YOLOv8s in most detection categories. Under high-intensity illumination conditions, distant pixels are frequently surrounded by saturated light regions, resulting in diminished contrast

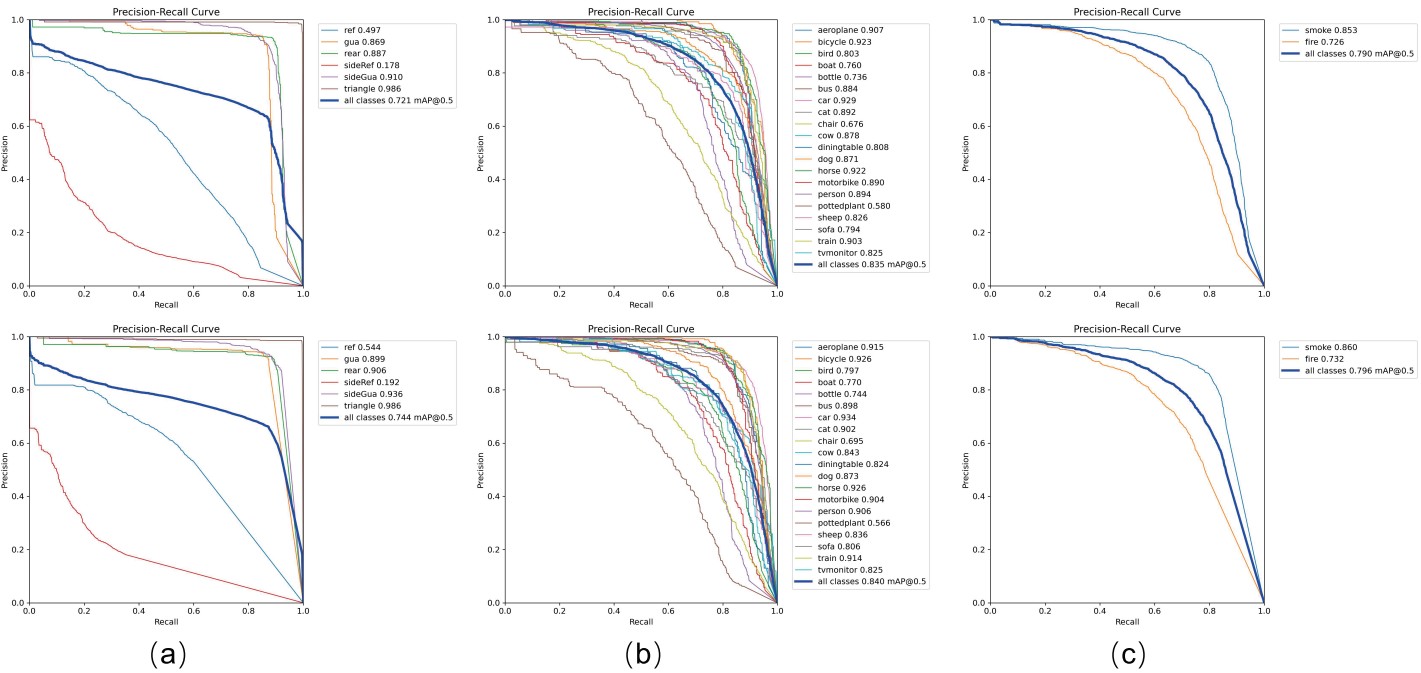

**Figure 9 The precision-recall (P-R) curves of YOLOv8s and VBM-YOLO across different datasets.** The upper section shows YOLOv8s, while the lower section corresponds to VBM-YOLO. As shown in the figure, (A–C) display the P-R curves for the Vehicle Body Marker dataset, the PASCAL VOC dataset, and the D-Fire dataset, respectively. In the P-R curves, various vehicle body markers are annotated as follows: rear reflective markers (ref), side reflective markers (sideRef), rear guard devices (gua), side guard devices (sideGua), rear markers (rear), and triangular markers (triangle).

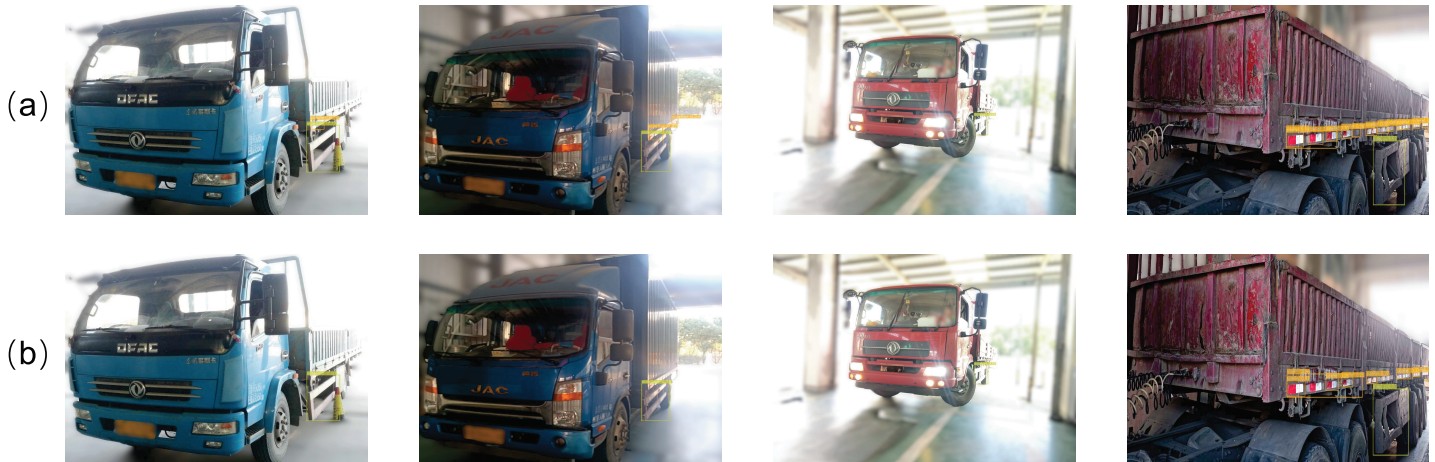

**Figure 10 A comparison of the detection performance between VBM-YOLO and YOLOv8s.** As shown in the figure, (A) and (B) represent the tests conducted under strong lighting conditions and at varying distances, respectively.

differentiation between adjacent pixels. This photometric interference may cause detection models to overlook subtle variations between pixels during feature extraction or even completely ignore the feature information from these light-affected regions. To address this limitation, our detection framework incorporated CSCA module that explicitly

**Table 2 Comparison of different object detection models on the PASCAL VOC dataset.**

| Model | mAP@0.5 | mAP@0.5:0.95 | Params (M) | GFLOPs | FPS |
|---|---|---|---|---|---|
| YOLOv5s | 0.786 | 0.532 | 7.0 | 15.8 | 143 |
| YOLOv6s | 0.837 | 0.645 | 16.3 | 45.3 | 163 |
| YOLOv7-tiny | 0.79 | 0.533 | 6.0 | 13.1 | 165 |
| YOLOv8s | 0.835 | 0.637 | 11.1 | 28.4 | 147 |
| RT-DETR-R18 | 0.837 | 0.612 | 20.1 | 60.0 | 86 |
| YOLOv9-C | 0.884 | 0.709 | 25.4 | 102.1 | 103 |
| YOLOv10-M | 0.858 | 0.674 | 16.4 | 59.1 | 112 |
| YOLOv11-S | 0.793 | 0.591 | 9.4 | 21.5 | 110 |
| YOLOv12-S | 0.81 | 0.61 | 9.3 | 21.4 | 80 |
| VBM-YOLO | 0.84 | 0.648 | 18.3 | 44.9 | 138 |

**Table 3 Comparison of different object detection models on the D-Fire dataset.**

| Model | mAP@0.5 | mAP@0.5:0.95 | Params (M) | GFLOPs | FPS |
|---|---|---|---|---|---|
| YOLOv5s | 0.78 | 0.414 | 7.0 | 15.8 | 143 |
| YOLOv6s | 0.778 | 0.46 | 16.2 | 45.3 | 165 |
| YOLOv7-tiny | 0.776 | 0.494 | 6.0 | 13.1 | 168 |
| YOLOv8s | 0.79 | 0.474 | 11.1 | 28.4 | 145 |
| RT-DETR-R18 | 0.586 | 0.324 | 20.0 | 60.0 | 88 |
| YOLOv9-C | 0.802 | 0.491 | 25.4 | 102.1 | 105 |
| YOLOv10-M | 0.785 | 0.468 | 16.4 | 59.1 | 112 |
| YOLOv11-S | 0.791 | 0.472 | 9.4 | 21.5 | 110 |
| YOLOv12-S | 0.794 | 0.476 | 9.3 | 21.4 | 82 |
| VBM-YOLO | 0.796 | 0.494 | 18.2 | 44.9 | 139 |

preserves illumination-aware feature distinctions between proximal and distant pixels throughout the feature hierarchy. As illustrated in Fig. 10, this mechanism achieved significant enhancement in the model's detection performance.

To further validate VBM-YOLO's generalization performance, we conducted comparison experiments on the public PASCAL VOC and D-Fire datasets. As shown in Tables 2 and 3, comparison experiments were conducted on the publicly available PASCAL VOC and D-fire datasets to evaluate the generalization capabilities of our proposed model against other mainstream object detection models.

For results in the PASCAL VOC dataset, though our VBM-YOLO model is specifically designed for detecting vehicle body markers, it also demonstrated superior performance, surpassing the baseline YOLOv8s model and other earlier YOLO versions. VBM-YOLO achieved higher accuracy than RT-DETR, which is based on the ResNet18 backbone, despite having fewer parameters. While YOLOv9-C showed higher accuracy than VBM-YOLO, it required 7.1 million additional parameters.

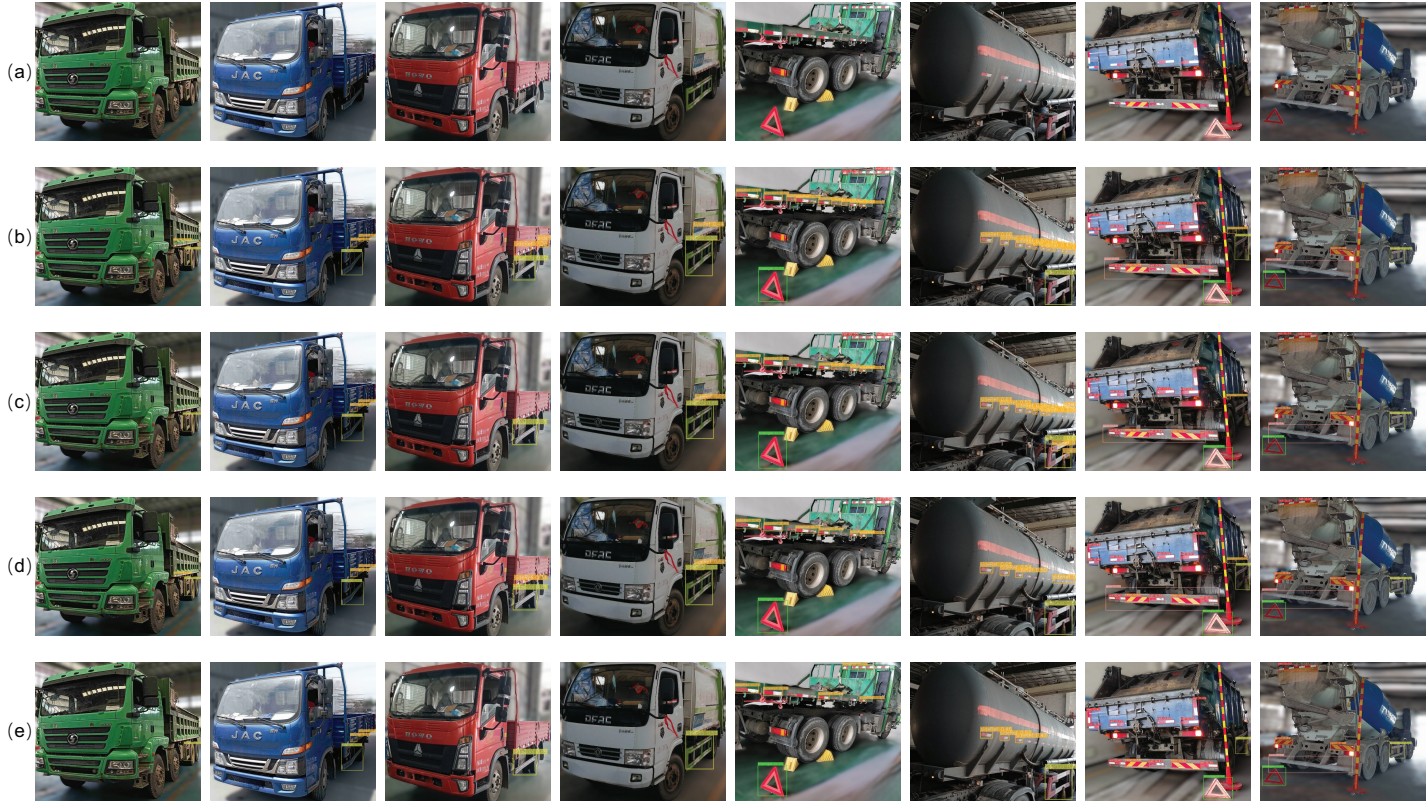

**Figure 11 Demonstration of detection performance across different models.** As shown in the figure, (A) represents the original image, while (B–E) illustrate the different detection results of the VBM-YOLO model, YOLOv8s, YOLOv9-C, and YOLOv10-M, respectively. Different vehicle body markers are labeled in the images as follows: rear reflective sign (ref), side reflective sign (sideRef), rear guard (gua), side guard (sideGua), rear sign (rear), and triangle sign (triangle).

For results in the D-fire dataset, while YOLOv10-M achieved higher accuracy than VBM-YOLO on the PASCAL VOC dataset, its accuracy on the D-fire dataset was significantly lower than that of VBM-YOLO. Furthermore, while YOLOv9-C had a higher mAP@0.5, VBM-YOLO achieved a higher mAP@50:95 with fewer parameters. Excluding YOLOv9-C, which has substantially more parameters than VBM-YOLO, our model achieved the highest accuracy on both metrics in the D-fire dataset compared to other models.

The above analysis indicates that the proposed VBM-YOLO model not only excels in vehicle body markers detection but also exhibits strong generalization capabilities across other detection tasks. To visually demonstrate the performance advantages of our VBM-YOLO algorithm, Fig. 11 compares the detection capabilities of VBM-YOLO, YOLOv8, YOLOv9, and YOLOv10. Although there are significant differences in the number of model parameters between YOLOv8 and YOLOv9, their final detection performances do not exhibit notable distinctions. The recently released YOLOv10 has made substantial progress in mitigating latency issues. However, its specific optimizations for the COCO dataset and lightweight design may somewhat limit its efficacy in vehicle body markers detection. Notably, all three models exhibit a certain degree of missed

**Table 4  Ablation study of main components of the VBM-YOLO on the vehicle body markers dataset.**

| CSCA | MSSFPN | ASFF-softmax | ASFF-sigmoid | C2f | C2f-Rep | AGB | mAP@0.5 | mAP@0.5:0.95 |
|------|--------|--------------|--------------|-----|---------|-----|---------|--------------|
|      |        |              |              |     |         |     | 0.721   | 0.428        |
| ✓    |        |              |              |     |         |     | 0.731   | 0.432        |
|      | ✓      | ✓            |              | ✓   |         |     | 0.741   | 0.461        |
|      | ✓      |              | ✓            | ✓   |         |     | 0.742   | 0.462        |
|      | ✓      |              | ✓            |     | ✓       |     | 0.741   | 0.465        |
| ✓    | ✓      |              | ✓            |     | ✓       |     | 0.745   | 0.467        |
| ✓    | ✓      |              |              |     | ✓       | ✓   | 0.744   | 0.471        |

detections. VBM-YOLO, benefiting from techniques designed to address information loss, demonstrates a significant advantage in reducing missed detections compared to the other algorithms.

## Ablation experiments

The ablation experiments were conducted to verify the effectiveness of each improvement module in the VBM-YOLO model on the vehicle body markers dataset. The results are shown in Table 4. Initially, the CSCA module was added individually before the SPPF module in the baseline model. It was observed that mAP@0.5 and mAP@0.5:0.95 increased by 1% and 0.4%, respectively, compared to the baseline model. Subsequently, a separate ablation experiment was conducted on the MSSFPN, replacing the original baseline model's neck network. This led to significant improvements, with mAP@0.5 and mAP@0.5:0.95 increasing by 2% and 3.3%, respectively. These results demonstrate the effectiveness of the proposed MSSFPN feature pyramid architecture for feature fusion and extraction.

Additionally, we found that employing the ASFF module with a sigmoid function in MSSFPN produced better results than using the ASFF module with a softmax function. Both mAP@0.5 and mAP@0.5:0.95 increased by 0.1%. An ablation experiment was also conducted on the C2f module within the MSSFPN. It was found that using the C2f-Rep model, which incorporates channel fusion feature information, resulted in a 0.1% decrease in mAP@0.5 but a 0.3% increase in mAP@0.5:0.95. Therefore, the MSSFPN was implemented with the ASSF module using the sigmoid activation function and the C2f-Rep module.

To explore the effect of using the CSCA module and MSSFPN simultaneously, the CSCA attention mechanism was added before each target layer in the VBM-YOLO model's backbone, and the MSSFPN was used in the neck network. A significant improvement in accuracy was observed compared to using the CSCA and MSSFPN individually. Compared to the initial baseline model, mAP@0.5 and mAP@0.5:0.95 increased by 2.4% and 3.9%, respectively. Furthermore, to alleviate the loss of gradient information for each target layer during backpropagation, the AGB branch was used to supervise training and removed during inference. It was observed that although mAP@0.5

**Table 5 Performance of different attention mechanisms in the VBM-YOLO model on the vehicle body markers dataset.**

| Attention mechanism | mAP@0.5 | mAP@0.5:0.95 | GFLOPs |
|---|---|---|---|
| CSCA | 0.745 | 0.467 | 45.2 |
| iRMB (*Zhang et al., 2023*) | 0.739 | 0.465 | 75.9 |
| Biformer (*Zhu et al., 2023*) | 0.738 | 0.464 | 89.1 |
| EMA | 0.722 | 0.434 | 34.8 |
| DilateFormer (*Jiao et al., 2023*) | 0.719 | 0.432 | 37.3 |

**Figure 12 Performance comparison of different attention modules.** As shown in the figure, (A) and (B) represent the heatmaps of CSCA, iRMB, Biformer, EMA, and DilateFormer on two different images.

decreased by 0.1% compared to not using AGB supervision, mAP@0.5:0.95 increased by 0.4%. Therefore, we decided to use the AGB branch to assist in training our model.

In addition, to further validate the superiority of the proposed CSCA attention mechanism over other widely used attention mechanisms in vehicle body markers detection, we replaced CSCA in the VBM-YOLO with attention mechanisms employing similar approaches, such as iRMB (*Zhang et al., 2023*), Biformer (*Zhu et al., 2023*), EMA (*Ouyang et al., 2023*), and DilateFormer (*Jiao et al., 2023*). It is important to note that the AGB branch was not used during these experiments. The results, as shown in Table 5, clearly demonstrate that our attention mechanism achieves the best balance between accuracy and computational resources when incorporated into the model, compared to other attention modules.

We used heatmaps, as illustrated in Fig. 12, to visually observe the performance of each attention mechanism. Compared to our CSCA attention mechanism, the heatmaps for other attention mechanisms exhibit missing or lighter thermal information in the regions requiring detection. In contrast, the CSCA heatmap shows significantly darker color distribution in the areas that need to be detected. These results indicate that the CSCA attention module captures key information in the features more effectively than other attention modules.

## CONCLUSION

In this study, we propose an improved vehicle body markers detection algorithm, VBM-YOLO, based on YOLOv8s, which addresses the limitations of current mainstream object detection models in this specific application. In the VBM-YOLO model, we utilized the proposed CSCA, MSSFPN, and AGB to reduce information loss. Experimental results demonstrate that our algorithm achieves superior performance on the vehicle body markers dataset with an mAP@50:95 of 47.1%, outperforming YOLOv8 by 4.3%, YOLOv9-C by 2.4%, YOLOv10-M by 5.1%, YOLOv11-M by 5%, and YOLOv12-M by 3.6%, while maintaining an effective balance between accuracy and computational resource consumption. Additional evaluations on PASCAL VOC and D-Fire datasets confirm enhanced detection performance compared to the baseline model, demonstrating improved generalization capabilities. As our model is specifically designed for vehicle body markers detection rather than general object detection, its performance relative to universal detection models remains suboptimal. Future research will employ knowledge distillation and incremental learning techniques to enhance generalization capabilities. Under high-intensity illumination conditions, although our model enhances performance by computing differences between distant and proximal pixels, subtle variations might be overlooked when the computed values are uniformly low across the entire distribution. Future work will explore adaptive amplification of proximal light-source gradients through enhanced histogram equalization techniques, including contrast-limited adaptive histogram equalization (CLAHE) and learnable attention-based enhancement operators. Additionally, we aim to develop compact network variants optimized for small-scale training scenarios without compromising detection accuracy. Pending successful optimization, we intend to deploy the model in vehicle inspection stations to automate compliance checks of vehicle markings, thereby enhancing road safety regulations enforcement.

### Funding
The authors received no funding for this work.

### Competing Interests
The authors declare that they have no competing interests.

### Author Contributions
- Bin Wang conceived and designed the experiments, performed the experiments, analyzed the data, performed the computation work, prepared figures and/or tables, authored or reviewed drafts of the article, and approved the final draft.
- Chao Li conceived and designed the experiments, prepared figures and/or tables, authored or reviewed drafts of the article, and approved the final draft.
- Chao Zhou performed the experiments, prepared figures and/or tables, authored or reviewed drafts of the article, and approved the final draft.

- Jun Sun analyzed the data, prepared figures and/or tables, authored or reviewed drafts of the article, and approved the final draft.

## Data Availability

The code is available at GitHub and Zenodo:

- https://github.com/wwwwbsiu/VBM-YOLO

- wwwwbsiu. (2024). wwwwbsiu/VBM-YOLO: VBM-YOLO: An enhanced YOLO model with reduced information loss for vehicle body markers detection (main). Zenodo. https://doi.org/10.5281/zenodo.14264171.

The third party datasets are available at:

- PASCAL VOC: http://host.robots.ox.ac.uk/pascal/VOC/voc2012/index.html#devkit.

- D-Fire: https://github.com/gaiasd/DFireDataset.

- Vehicle body markers dataset: Wang, B. (2024). vehicle body markers dataset [Data set]. Zenodo. https://doi.org/10.5281/zenodo.14264265.

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
