# Peer review of "VBM-YOLO: an enhanced YOLO model with reduced information loss for vehicle body markers detection"

_PeerJ Computer Science, doi:10.7717/peerj-cs.2932_

## Round 0.1 · original submission · Minor Revisions

Both reviewers have requested minor revisions.

·

Basic reporting

The introduction provides a strong motivation for the study, highlighting the significance of vehicle body markers detection for traffic safety. They proposed a YOLO Varianr for traffic safety analysis, they curated an in-house dataset.

The manuscript is well-written in professional and clear English.
There are minor grammatical inconsistencies (e.g., unnecessary repetitions and awkward phrasing in some sections). While they do not hinder comprehension, proofreading would enhance readability.
Example: In the introduction, the phrase "identifying body markers on medium and large vehicles, such as reflective signs and guards, is critical to ensuring safe road travel" could be restructured for clarity.
Suggestion: The authors may consider a professional language editing service or review by a native English speaker for polishing.

Experimental design

The study demonstrates that VBM-YOLO improves accuracy, but it does not explicitly discuss the computational cost in terms of FLOPs or inference speed on different hardware.
Suggestion: Include a comparison table or discussion of inference time (e.g., FPS on GPU/CPU) to help readers understand the trade-off between accuracy and computational efficiency.

Validity of the findings

Failure Case Analysis

The paper does not discuss cases where the model underperforms, such as extreme lighting conditions or occlusions.

Suggestion: Provide a brief failure analysis with examples to improve transparency and guide future improvements.


Dataset Preprocessing Details

The dataset section lacks details on preprocessing techniques such as augmentation, normalization, or balancing.

Suggestion: Describe any preprocessing steps taken to enhance model generalization.

Reviewer 2 ·

Basic reporting

The manuscript is written in clear, professional English, adhering to academic standards. The technical terminology is appropriate, and the structure is logical, with a well-defined introduction, methodology, experiments, and conclusion.

The introduction effectively contextualizes the importance of vehicle body marker detection for road safety and highlights the limitations of existing YOLO models. The literature review is comprehensive, covering recent advancements in object detection and attention mechanisms. However, the introduction could benefit from a more explicit statement of the knowledge gap being addressed. For instance, while the authors mention the lack of models specifically designed for vehicle body markers detection, they could elaborate on why existing models (e.g., YOLOv8, YOLOv9) are insufficient for this task.

The paper follows a standard structure, with clear subsections. Figures are well-labeled and relevant, particularly the visualizations of the proposed CSCA mechanism and MSSFPN architecture.

The authors have provided access to the datasets and code, which is commendable. However, the metadata for the datasets could be more detailed to facilitate replication by other researchers. For example, including information on the distribution of lighting conditions, distances, and angles in the vehicle body markers dataset would be beneficial.

Experimental design

The research question is well-defined: to improve the detection of vehicle body markers by addressing feature and gradient information loss in YOLO models.

The methodology is rigorous and well-documented. The proposed enhancements CSCA, MSSFPN, and AGB are innovative and address specific limitations of existing models. The authors provide sufficient detail to replicate the experiments, including hyperparameters, training settings, and evaluation metrics. However, the paper could benefit from a more detailed discussion of the tradeoffs between accuracy and computational efficiency, particularly in the context of realtime applications.

The hyperparameters (e.g., batch size, learning rate, optimizer) are clearly stated, and the use of SGD with momentum is justified. However, the authors should consider including a hyperparameter sensitivity analysis to demonstrate the robustness of their model to different settings.

Validity of the findings

The experiments are well-designed, with comparisons to state-of-the-art models (e.g., YOLOv8, YOLOv9, RT-DETR) on multiple datasets. The results are statistically sound, with improvements in [email protected] and [email protected]:0.95 clearly demonstrated. However, the authors should provide confidence intervals or statistical tests to validate the significance of the improvements.

The model’s generalization is tested on public datasets (PASCAL VOC and D-Fire), which is a strength. However, the performance on these datasets is only marginally better than baseline models, suggesting that the proposed enhancements are highly task-specific. The authors should discuss this limitation and explore ways to improve generalization further.

Additional comments

The paper presents a novel and well-executed approach to improving YOLO models for vehicle body markers detection. The methodology is sound, and the results are promising. However, the limited generalization performance and lack of discussion on computational overhead are significant concerns. I recommend minor revisions to address these issues before acceptance.

Please also consider:
Adding a real-time performance analysis to demonstrate the model’s suitability for industrial applications.
Include a broader comparison with other state-of-the-art models to further validate the proposed approach.
Provide a more detailed discussion of the practical implications of the proposed model for road safety and autonomous driving.

---

## Round 0.2 · accepted · Accept

For vehicle safety detection to work, it is crucial to correctly identify body marks on big and medium-sized cars. To fix the problems with the earlier YOLO series' feature and gradient information loss, a new model called VBM-YOLO, which stands for Vehicle Body Markers, has been developed.